# How Important Are Genes to Achieve Longevity?

**DOI:** 10.3390/ijms23105635

**Published:** 2022-05-18

**Authors:** Calogero Caruso, Mattia Emanuela Ligotti, Giulia Accardi, Anna Aiello, Giovanni Duro, Damiano Galimberti, Giuseppina Candore

**Affiliations:** 1Laboratorio di Immunopatologia e Immunosenescenza, Dipartimento di Biomedicina, Neuroscienze e Diagnostica Avanzata, Università di Palermo, 90133 Palermo, Italy; mattiaemanuela.ligotti@unipa.it (M.E.L.); giulia.accardi@unipa.it (G.A.); anna.aiello@unipa.it (A.A.); giuseppina.candore@unipa.it (G.C.); 2Istituto per la Ricerca e l’Innovazione Biomedica, Consiglio Nazionale delle Ricerche, 90146 Palermo, Italy; giovanni.duro@irib.cnr.it; 3Associazione Italiana Medici Anti-Aging, 20133 Milan, Italy; damiano.galimberti@gmail.com

**Keywords:** APOE, FOXO3A, genes, GWAS, longevity, long-life families, SNP, survival

## Abstract

Several studies on the genetics of longevity have been reviewed in this paper. The results show that, despite efforts and new technologies, only two genes, APOE and FOXO3A, involved in the protection of cardiovascular diseases, have been shown to be associated with longevity in nearly all studies. This happens because the genetic determinants of longevity are dynamic and depend on the environmental history of a given population. In fact, population-specific genes are thought to play a greater role in the attainment of longevity than those shared between different populations. Hence, it is not surprising that GWAS replicated associations of common variants with longevity have been few, if any, as these studies pool together different populations. An alternative way might be the study of long-life families. This type of approach is proving to be an ideal resource for uncovering protective alleles and associated biological signatures for healthy aging phenotypes and exceptional longevity.

## 1. Introduction

The aging process is driven by the accumulation of molecular damage, resulting in a gradual increase in the fraction of cells and organs carrying defects. The age-related increase in damage levels interferes with both the performance and the functional reserves of tissues and organs, resulting in a disruption of the self-organization system and a reduced capacity to adapt to the environment. Frailty, disability, age-related diseases, and, ultimately, death ensue. Maintenance mechanisms slow down the damage accumulation rate. These mechanisms are positively or negatively modulated by various factors, i.e., genetics, epigenetics, sex and gender, socioeconomic and educational status, chance and circumstances of life, nutrition and physical activity, stress management and social support, and pathogenic load. Different combinations of these factors create the possibility of avoiding, delaying, or controlling age-related diseases [1,2].

Longevity should be considered a specific country/population concept, as different populations/countries, due to different historical, anthropological, socioeconomic, and environmental factors, show great variability in their life expectancy. In “absolute” terms, however, the definition of longevity is based on the maximum duration of life of human beings. The canonical age of 100 is considered the threshold of longevity, although people over 95 years are called long-lived individuals (LLIs). Supercentenarians are, instead, people who have reached the age of 110, while semi-supercentenarians are people between the ages of 105 and 109 [1,3,4].

Demographic evidence shows a continued decrease in mortality in old age as well as an increase in the maximum age at death, which might gradually extend human longevity. This would suggest that no strict constraints exist for human longevity. However, survival improvements with age tend to decline after age 100 and the age of death of the world’s oldest person (Jeanne Calment, 21 February 1875–4 August 1997 [4]) has not changed in the last 30 years. The analysis of the data of all Italians aged 105 or over between 2009 and 2015 (born in 1896–1910) has provided proof of the existence of a plateau of mortality at extreme age. These studies suggest that the maximum human life span is fixed and subject to natural constraints, likely linked to the laws of physics [2,5,6].

Longevity is not a matter of genes. A few years ago, this message appeared in all the newspapers of the world, following the publication of a large study that dissected the genealogical trees of 400 million individuals, tracing back generations, and including dates of birth, death, places, and family ties. According to this study, as well as another published the same year, genes would play a very marginal role in achieving longevity [7,8]. However, these extensive studies have analyzed the influence of genetics in terms of lifespan, but not in terms of longevity [9]. Both articles refer, indeed, to a low prediction of descendant longevity based on the ages reached by their ancestors/parents. Despite that no age cut off was adopted and all ages were included, it is reasonable that the ages of individuals used in these studies reflect the average population lifespan, which includes only a small percentage of centenarians (less than 1 in 5000). This would explain the low heritability observed of the longevity trait. On the other hand, several years ago, Perls et al. [10,11] clearly showed a high heritability of the exceptional longevity phenotype that becomes even higher as the ages of the study participants increase (see also paragraph 4).

There is no doubt that human beings are the result of their genes, but they would not be the same without the experiences that they have had since leaving the mother’s womb. As an example, in cohorts of U.S. births between 1918 and 1919, prenatal exposure to the influenza pandemic (Influenza A, H1N1 subtypes) was shown to be associated with ≥20% excess of cardiovascular diseases (CVDs) at 60–82 years of age, compared with cohorts born without exposure to the epidemic. The height of adults at the time of enrolment for World War II was lower than that of people born in adjacent years, and school performance was also lower. Therefore, prenatal exposure to maternal flu, even uncomplicated, can have consequences on future extrauterine life, likely through epigenetics [12]. It cannot be ruled out that these epigenetic changes might also be due to stress as a result of the consequences of war. On the other hand, everyone would not be the same if born in another part of the world as clearly demonstrated by Table 1.

Therefore, longevity is reached more easily in “rich” countries. The “Lifepath” research consortium investigated the effects of socioeconomic inequalities on the biology of successful aging. Results suggest that unfavorable socioeconomic conditions from birth affect biological systems from molecules to organs. These data have important implications for policy, suggesting that addressing unfavorable socioeconomic conditions, including education level, is as important as considering well-known risk factors, such as tobacco and alcohol use, low fruit and vegetable consumption, obesity, and a sedentary life, and that the effects of preventive interventions in the first years of life must integrate those in adulthood [14].

However, this belief is apparently disproved by the case of the Okinawan population, as pointed out by Cockerham et al. [15]. Okinawa is one of the four classic blue zones, which are limited regions where the population shares a common lifestyle and environment, and whose exceptional longevity, superior to that of the rest of the country, has been carefully verified. It is seemingly paradoxical that the four classic blue zones are economically slightly underdeveloped compared with the rest of their respective countries. This paradox disappears if the economic development of these areas is evaluated in relative rather than in absolute terms. As an example, when, around the middle of the twentieth century, regions such as Sardinia and Ikaria, two other blue zones, began their economic development, this involved the transition from extreme poverty to relative material wellbeing, with a reduced social stratification. The absence of a true class gradient and the consequent absence of social competition and individual stress could have created individual health conditions much more favorable than those operating in the most competitive areas of the country [16].

Random events, i.e., chance, also interact with genetic background. Chance can be defined as the occurrence of events in the absence of any obvious intention or cause. Therefore, it must be distinguished from life circumstances that are events or facts that cause or help to cause something to happen, as an example, the accidental death of a potential centenarian. Both immunological repertoire and the architecture of brain synapses, which influence survival, are also linked to many small casual events. Many genes are, indeed, transcribed in minimal amounts of mRNA per cell, which can cause large random fluctuations in biosynthesis. Genomic instability is another important source of inherent random variability, as shown in aged mice, which have a mutation rate of up to 104 per gene per cell. Somatic mutations are random events since they result from mispairing, originating from the equilibrium that exists in solution between the tautomeric forms of the purine and pyrimidine bases. Epimutations can also occur through random changes in DNA methylation patterns, affecting gene expression [1,17,18].

Considering what is discussed above, asking whether longevity depends on environment or genetics is legitimate but oversimplified. In fact, it is necessary to consider everything that is important for longevity: luck (chance and circumstances of life), lifestyle (nutrition, physical activity, and environmental exposures, which also affect epigenetics), life experiences (pathogenic load, stress management and social support, and socioeconomic status and education), and biology, with sex and DNA (genetics and epigenetics) at the fore [1].

## 2. Study Methodologies

Longevity is a multifactorial trait for which gene–environment interactions as well as the complex interplay of multiple genes and pathways play a major role [19]. Yashin et al. [20] showed that longevity also depends on several small effect alleles. Genetic studies could lead to the identification of mechanisms that protect organisms from age-related diseases. Some of these mechanisms could be improved by specific environmental factors or lifestyle as discussed below for Apolipoprotein(APO)E and Forkhead box O3(FOXO3)A.

Over time, different genetic approaches have been adopted based on the available platforms. It started with a candidate gene approach as part of a case–control study design, followed by sibling pair linkage analysis, and returned to the case–control study with single nucleotide polymorphism (SNP) array, imputation, and, lastly, sequencing of the whole genome.

### 2.1. Genetic Association Studies

Genetic association studies analyze whether the allele, mostly identified as SNP, of a genetic variant is found more often than expected in individuals with the phenotype under study. The case–control study is a retrospective analysis that starts from two different groups, one with a disease/phenotypic trait and one without it. The aim of the study is to evaluate the presence of significant differences in the rate of exposure to a given risk factor (alleles in this case) between the two groups. The candidate gene approach focuses on associations between genetic variation within pre-selected genes of interest and phenotypes or disease states. Candidate genes are selected based on their presumed relevance to the disease or phenotypic trait in question. The fundamental unit for summarizing the size of associations is the odds ratio (OR). OR is the ratio of two probabilities, which in this context are the case probabilities for individuals who have a specific allele and the case probabilities for individuals who do not have the same allele. There are several criticisms of the candidate gene approach, since it has been shown that the candidate gene approach produces a high rate of false positive results [21,22].

For several years, genetic linkage analysis in siblings was the only tool available to highlight chromosomal regions that potentially harbor genetic variants that influence the phenotype under study. The approach that identifies excess allele sharing was initially performed using microsatellites as markers. DNA sequences that are close together on a chromosome, detected by a known marker, are inherited together during the meiosis phase of sexual reproduction. However, for common diseases and complex traits, the results of genetic linkage studies have proved difficult to reproduce. While linkage analysis was successfully used to identify genetic variants that contribute to rare disorders, it did not perform that well when applied to multifactorial traits such as longevity [21,22].

### 2.2. Genome-Wide Association Study

A genome-wide association study (GWAS) is an investigation of all, or nearly all, of the genes of different individuals to determine gene variations between individuals under consideration. An attempt is made to associate the observed differences with some traits, in this case longevity. GWAS studies allow researchers to sample one million or more SNPs from each subject, evenly distributed across the genome. Each of these SNPs it is then analyzed to understand if the SNP frequency is significantly different between the case and the control groups. GWAS are necessarily operated without starting from a hypothesis; the research is carried out on the whole genome rather than focusing on a small group of candidate genes. The results should be replicated in other samples of the same population, and many variables may be responsible for failure to replicate results such as differences in the age, gender, and health status of the participants of the two groups, younger and LLIs (see also Section 2.5). Furthermore, the GWAS approach is penalized from a statistical point of view, since the huge number of comparisons requires methods to correct for multiple testing, which means the adoption of very low *p* values of significance, to avoid the association with false positives related to chance. Except for the APOE locus, the results obtained with GWAS are not always replicable. This underscores the need for larger studies or an alternative study design to discover common polymorphisms with minor genetic effects and rare variants with high penetrance that affect longevity. As for the power to capture true association in GWAS, a cohort of thousands of individuals is needed to identify a sufficient OR [21,22].

To improve the power of GWAS studies, meta-analysis has become a common tool in genetic epidemiology to accrue large sample sizes. Sebastiani et al. [23] described how meta-analysis approaches applied to the study of the genetics of human longevity appear to have several limitations. An extended definition of LLIs, indeed, understood as subjects who survived up to 85 years and beyond, used to increase the size of the sample to be studied through a meta-analysis, inevitably increases the heterogeneity of the phenotypes. The most important problem is represented by the heterogeneity of the study population when different cohorts are analyzed in the same study to increase the statistical power that depends on the sample size (see Section 2.4).

### 2.3. Next Generation Sequencing

New cheap and fast genome sequencing methods might provide a realistic alternative to GWA studies. High-throughput sequencing, next generation sequencing (NGS), has, indeed, the potential to bypass some of the shortcomings of GWAS. The availability of this huge amount of data does not correspond to a simpler and more efficient way to discover genetic variants associated with the phenotype. To achieve adequate statistical power, NGS studies require very large case and control populations, due to the large number of genetic variants and rare variants [22].

### 2.4. Other Possible Approaches

Another possible approach is represented by the study of centenarian offspring (CO). They are typically in their 70s and 80s and display a lower prevalence of all-cause mortality, cancer, diabetes, and CVDs, suggesting that their resilience against disease and death may be at least partly genetic [24]. However, the most important information that this model can offer concerns the role of the immune system in achieving longevity. The potential role of the immune system in achieving one hundred or more years of age is seemingly questioned by the fact that even centenarians exhibit the same age-related changes in T and B lymphocytes observed in older people (however, they show a well-preserved cytotoxic activity of NK cells) [25]. It must be considered that their acquired immune system has been subjected to a pathogenic burden for decades not predicted by evolution. To gain insight into the role of the immune system in longevity, a better model is represented by the offspring. In fact, the immune systems of offspring have been subjected to an antigenic load for a shorter time and they have an appropriate control group important for this type of study, i.e., older people without centenarian parents. Accordingly, both B and T subsets of CO present an intermediate phenotype between old and younger people, showing a “younger” immune profile that might play a relevant role in making CO able to continue fighting off new infections, hence, prolonging their life [24,26].

A further alternative method could be that of the study of long-life families (LLF). This approach is proving to be an ideal resource for uncovering protective alleles and associated biological signatures for healthy aging (HAP) and exceptional longevity phenotypes. Ongoing studies are already showing how HAP LLFs are inheritable, cross-sectionally and longitudinally, and how these families are heterogeneously protected. They are also finding evidence that multiple, rare, and protective variants likely drive some HAPs and longitudinal trajectories [27].

Over the last few years, a lot of GWAS data have accumulated, even using alternative phenotypes (paternal longevity, health span, or some meta-analysis on these data), as described below. Six European-ancestry GWAS of human aging traits, i.e., health span, father and mother lifespan, exceptional longevity, frailty index, and self-rated health, have been combined in a principal component framework to maximize their shared genetic architecture [28]. In another study, authors have combined existing genome-wide association summary statistics for health span, parental lifespan, and longevity in a multivariate framework [29]. The challenge of studying aging genetics in humans, low heritability, and limited samples, can be overcome to some extent by combining large studies of closely related phenotypes that capture elements of the aging process. In a further study, the DNA of over 500,000 people was analyzed to reveal the specific “genetic fingerprints” of each participant and results were related to the parental lifespan [30].

### 2.5. Limits of These Studies

To pool different populations allows the identification of genes and pathways that are important for longevity and healthy aging if shared among the different populations examined. However, the “ecological” dimension of healthy aging and longevity is lost. It should be borne in mind, indeed, that the genetic determinants of longevity are dynamic and depend on the environmental history of a given population. Genes specific for a given population are in fact believed to play a more important role than those shared between different populations [19,31,32]. This occurs because gene–environment interactions are specific for a given population, due to the variability of environmental and cultural contexts such as, among others, food habits and lifestyle (see also Section 3 below). The choice of controls is another debated point; however, it is sufficient to use unrelated samples of the general population matched by geographic origin, as the prevalence in the control group of individuals who will become LLIs is negligible due to the rarity of the trait.

## 3. Genes Shown to Be Associated with Longevity

Centenarians are the best example of extreme longevity, representing selected individuals in which the appearance of major age-related diseases has been consistently delayed or avoided. There is growing evidence that the genetic component of longevity becomes higher with survival at the age of over 90 years. Conceptually, longevity should correlate either with the presence of protective alleles or the absence of detrimental alleles [22,33,34]. However, the presence of detrimental alleles does not compromise the achievement of longevity. Among the various explanations of this fact are: (1) the opposing influence of genes on different health traits; (2) the antagonistic change of gene effect at different ages (antagonistic pleiotropy); (3) the gene–gene interaction (epistasis); and (4) the gene–environment interaction [20,35,36,37]. However, there are only two genes whose variants have been almost consistently associated with longevity: APOE and, to less extent, FOXO3A [21,22,34].

### 3.1. APOE

The APOE protein is the principal cholesterol carrier that drives lipid transport and injury repair in the brain. APOE genetics polymorphisms are determined by three alleles, ε2, ε3, and ε4, defined by combinations of genotypes of the SNPs rs7412 and rs429358. The products of the three alleles differ in several functional properties. APOEε3 is the most frequent allele in all human groups. Total APOE levels in plasma in very old individuals were found to be associated with lower total cholesterol and LDL cholesterol levels, which, in turn, were associated with the APOEε2 allele [38,39]. Genetic variations in APOE are well known to be associated with longevity and lifespan. In the first report more than two decades ago, in a small candidate gene study, Schachter et al. [40] showed that French centenarians had a very low frequency of APOε4 allele; in addition, an increased frequency of the allele APOEε2 in centenarians was observed. Since then, there have been numerous candidate gene studies and GWAS studies, including individuals of diverse ancestry, which have confirmed associations of APOE with longevity. The APOE ε2 and ε4 variants have previously been associated with a decreased (ε2) or increased (ε4) risk for several age-related diseases, such as CVDs and Alzheimer’s disease (AD), which could explain the APOE effect on longevity [29,38,39,41,42].

The APOEε4 allele is the ancestral proinflammatory allele, joined by APOEε3 and then APOEε2 in the human species. Under conditions of several infections, uncertain food, and shorter life expectancy, APOEε4 may be adaptive for reducing mortality. As evidence for this hypothesis, APOEε4 carriers have less severe liver damage during hepatitis C infections. The frequency of the APOEε4 allele remained higher in countries where food supply is, or was until the recent past, scarce, or sporadically available. In fact, this allele is linked to elevated cholesterol blood levels [43,44,45]. It is a “thrifty” allele. The term “thrifty alleles” was originally coined to indicate alleles that enable individuals to efficiently collect and process food to deposit fat during periods of food abundance to provide for periods of food shortage as feast and famine. In this way, the persistence of genes favoring the onset of type 2 diabetes (T2D) was explained since they would have been advantageous for hunter–gatherer populations [46]. As human lifespan lengthened and cognitive and cardiovascular health became more important, the APOEε3 allele spread, while the APOEε4 allele was maintained in all populations by balancing selection. The exposure of people carrying APOEε4 to the new affluent environmental conditions (Western diet and longer lifespans) could have rendered them susceptible to CVDs and AD [47].

In Sicilian LLIs, we did not observe the positive association of APOEε2 nor the negative one of APOε4 with longevity although this might be due to the small size of the sample [48]. These results are consistent with a recent analysis that shows that in South Italy there is a weaker protective effect of APOEε2 and no detrimental effect of APOEε4 [38]. In that study, the authors provided evidence that the genetic effect of APOE alleles changes based both on the country of residence and on genetic ancestry, suggesting the presence of environmental risk factors of a place of residence that modify the genetic effects of APOE. There was no deleterious effect of APOEε4 in subjects with Southern Italian ancestry living in the south of Italy; however, there was deleterious effect in those living in the U.S. These results suggest that factors related to living in the south of Italy may mitigate the deleterious effect of APOEε4. In particular, the Mediterranean diet (MedDiet) followed in Italy at a young age by the generations under study [48] should contribute to that difference. Detrimental effects of APOEε4 may be, indeed, alleviated through diet interventions, specifically the MedDiet [49,50]. The results are consistent with previous findings showing that the MedDiet reduces the risk of AD [51].

### 3.2. FOXO3A

The FOXO3A SNPs, in particular the rs2802292 G-allele (G > T), are other variants associated with longevity across many populations although gender-specific associations have been found in males, as in the Southern Italian Centenarian Study [52]. Interestingly, the FOXO3 rs2802292 G-allele has protective effects on several age-related diseases, in particular CVDs [53], but also cancer and bone fractures, and it is associated with better self-rated health, which strongly predicts mortality [54]. A meta-analysis of over 7900 cases and 9500 controls confirmed the association of the G allele of the SNP rs2802292 with longevity, especially in men [53]. This datum confirms and extends the results of a previous one, including the sex-specific differences in the association of a genetic variation with survival during old age [55]. This is not surprising because it has been claimed that men and women follow different strategies to attain longevity and several association studies show positive results only in men [56,57]. The reason is obviously multifactorial, with a sociocultural component that can be distinguished from biological trait linked to longevity, although in this specific case a direct effect of estrogen on the modulation of transcription might be relevant [54].

FOXO3A, which is part of the nutrient sensing pathway linked to insulin and insulin growth factor(IGF)-1, has an important role as “gatekeeper” by balancing the cell response to oxidative stress and nutrient availability. FOXO3A acts as a transcription factor on multiple homeostatic genes in response to decreased insulin/IGF-1 signaling. The SNP may improve the ability of FOXO3A in fighting oxidative stress by enhancing its interconnections with up- and down-stream molecular partners. Several studies in models have shown that modifications that impact on these signals are able to postpone aging as this pathway regulates many aspects of cell homeostasis, from cell survival to proliferation [58]. Therefore, it is conceivable to speculate that hyper or hypo activation of this signaling pathway, due to genetic variants that lead to different expressions of homeostatic genes. Finally, an inverse relationship between a healthy life and FOXO3 expression has been observed; this might reflect the fact that healthy individuals have less oxidative damage and require less FOXO3 to mitigate this damage [42]. Similarly, this finding could explain why in some long-lived populations the association with the allele involved in longevity was not found.

### 3.3. Other Genes

The GWAS Catalog [59] reports 59 studies showing 676 associations concerning health span, parental lifespan, or longevity. Overall, the genetic association studies reported as well as previous ones not carried out by GWAS suggest that the gene variants that contribute to a long life are involved in the basic maintenance and function of cells, tissues, and organs. Indeed, genes involved in DNA repair, telomere conservation, heat shock response, and the management of free radical levels have been found to contribute to longevity or, in the case of reduced functionality, to accelerate organism aging. Other genes associated with blood lipid levels, inflammation, and immunity as well as the cardiovascular systems have also been suggested to contribute to longevity because they reduce the risk of heart disease, stroke, and insulin resistance. In addition, as suggested by the studies in mice, the pathways involved in nutrient-sensing signaling and in regulating transcription have been shown to be involved in modulating human longevity [28,29,30,60,61,62].

In particular, the GWAS Catalog highlights studies with the largest initial sample size for specific experimental factor ontology as the paper by Timmers et al. [29]. The authors have combined existing GWA summary statistics for health span, parental lifespan, and longevity in a multivariate framework, increasing statistical power and identifying 10 genomic loci which influence all three phenotypes, of which five have not been reported previously at genome-wide significance. The majority of these 10 loci are associated with cardiovascular disease and some affect the expression of genes known to change their activity with age. In total, they implicated 78 genes, and found these to be enriched for aging pathways previously highlighted in model organisms, such as the response to DNA damage, apoptosis, and homeostasis.

Therefore, accumulation of DNA damage is associated with functional decline in the aging process. Thus, the maintenance of genomic integrity is a crucial factor for healthy life and longevity. Genomic instability is, indeed, considered a primary hallmark of aging [63]. Genome instability generally increases with age. DNA repair machinery controls genome stability [64]. Accordingly, several years ago Vijg and Suh suggested that longevity might be related to genomic integrity. The relatively low level of chromosomal aberrations observed in older persons should be a consequence of their genomic stability, and hence, a contributing factor to their attainment of advanced age [65]. The evidence for this is largely based on accelerated aging phenotypes of DNA repair in mice mutants and human progeroid syndromes [66,67].

It is intriguing that several other papers shown below, using different models and approaches, agree on the role of DNA repair (hence CVD control) in achieving longevity.

In a recent study [68], the results of whole genome sequencing of 81 Italian semi- and super-centenarians were presented and compared with a group of geographically matched healthy individuals. The methods used also allowed the analysis of somatic mutations. The results showed that extreme longevity was characterized by a peculiar genetic background likely responsible for an efficient DNA repair mechanism, as demonstrated by the low burden of somatic mutations such as that observed in younger controls. The genetic variants involved were replicated in a second cohort made of 333 Italian centenarians (>100 years) geographically matched to 358 controls (mean age: 60.7 ± 7.2). The low burden of somatic mutations might have contributed to protect them from CVDs. Accordingly, their existing polygenic risk score is not significantly different from those observed in younger controls. Thus, these subjects have escaped CVDs not because of a direct genetic protection toward cardiovascular risk but because they are protected from the burden of somatic mutations occurring during aging. These data support recent literature [69] that suggests a genetic signature in DNA repair mechanisms crucial for cellular homeostasis and protection from CVDs. This genetic background could therefore be at the basis of the extreme longevity of these subjects.

A very recent study [70], available as a preprint not yet formally peer-reviewed at the time of writing this paper, has identified a Sirtuin 6 (SIRT6) allele containing two linked substitutions (N308K/A313S) enriched in Ashkenazi Jewish centenarians as compared with controls. SIRT6 is an enzyme involved in multiple cellular pathways implicated in the regulation of aging and metabolism. Characterization of this SIRT6 allele demonstrated it to enhance stimulation of DNA double strand break repair. Additionally, it displayed a stronger interaction with Lamin A/C (LMNA), which might aid LMNA to organize/control nuclear protein–protein and protein–RNA interactions. Aberrant processing of LMNA results in human premature aging syndrome, Hutchison Gilford Progeria, while LMNA SNPs have been found to be associated with longevity. In addition, SIRT6 and its ortholog regulate lifespan in models [71,72,73,74,75].

The whole-genome sequencing of 208 colorectal crypts from 56 individuals provided insights into the somatic mutational landscape of 16 mammalian species. The somatic mutation rate per year varied greatly across species and showed a strong inverse relationship with species lifespan. Therefore, the most remarkable datum of this study is the inverse scaling of somatic mutation rates with lifespan, suggesting that mutation rates may be a contributing factor in aging and longevity [76].

Although these data require further study, it is known from both animal models and clinical data that progeroid syndromes [77] support the key role of improving genome maintenance in the attainment of longevity.

## 4. Conclusions

The importance of aging and longevity studies to address the medical, economic, and social problems associated with the increase in the number of non-autonomous old people suffering from invalidating diseases is linked to the extraordinary increase in the older population in the Western world. The increase in life expectancy with the consequent aging of the population is a great achievement of humanity, but it also represents a challenge that the Western world is currently facing, as aging is associated with increased susceptibility to many diseases such as CVDs, cancer, T2D, and AD. Therefore, it is necessary to fully understand the mechanisms of successful aging and longevity to prevent the harmful aspects of aging. Despite the efforts made by the international scientific community and the use of high-throughput genotyping methodologies, satisfactory results have not been obtained. The most significant associations have been obtained with the two genes, APOE and FOXO3A, which had already been identified for some time with simple case–control studies.

From the evolutionary point of view, longevity depends on the residual maintenance functions after the end of the reproduction period [78]. Aging depends on stochastic events and the aging phenotype is the result of the accumulation of cellular damage that cannot be repaired by the cellular maintenance systems that are running out [79]. Therefore, longevity depends on the possibility of survival after the end of the reproductive period and the genes that lead to longevity are “survival genes” rather than “longevity genes” [80].

Several studies of formal genetics strongly suggest the role of genes in achieving longevity. The comparison between the survival of the siblings of centenarians and that of their brothers-in-law, who likely shared the same lifestyle for most of their lives, showed that “the survival advantage” of siblings of long-lived subjects was not fully shared from their brothers-in-law. This suggested that beyond the family environment, there are genetic factors that influence survival and, consequently, longevity. This was not true comparing the survival of sisters with that of sisters-in-law. Interestingly, in this study, the survival curve of the sisters of long-lived subjects did not differ from the one of sisters-in-law, suggesting that the genetic component explains longevity in men more than in women [81]. The genetic component of lifespan in humans has also been analyzed by comparing the age of death of monozygotic and dizygotic twins. This has allowed to estimate that about 25% of the variation in human longevity can be due to genetic factors and indicated that this component is higher at older ages and is more important in males than in females [82,83,84].

It is thought that for the first eight decades of life, a correct lifestyle is a stronger determinant of health and life span than genetics. Genetics then appears to play a progressively important role in keeping individuals healthy and live as they age into their eighties and beyond. For centenarians, it reaches up to 33% for women and 48% for men. However, in general, the effect sizes were not large, suggesting that many genes of small effect play a role, as indeed in all multifactorial traits [33,53,61,82,85]; however, it needs to be considered that there is a dynamic interplay between genetic and environmental variations in the development of individual differences in health [86], and hence, longevity. Therefore, it is not surprising that GWAS-replicated associations of common variants with longevity have been few since they pool different populations losing the “ecological” dimension of longevity.

Overall, the findings discussed in this paper strongly suggest that longevity genetics are closely associated with protection against age-related diseases, particularly CVDs. The association with longevity is not surprising because CVDs are the leading cause of death globally, with an estimated 17.9 million deaths annually [87].

As previously stated, replicated associations of common variants with longevity have been few and the size of the effects are relatively modest. LLF studies are uniquely positioned to discover genetic and other factors related to longevity. It is now clear, indeed, that rare variants are numerous and that family studies show particular promise in their discovery [27].

## Figures and Tables

**Table 1 ijms-23-05635-t001:** Number of centenarians based on the income of the various countries.

	1980	2020
	per 10,000	
High-income countries	0.28	2.60
Middle-income countries	0.06	0.42
Low-income countries	0.01	0.03
Total	0.10	0.74

Similar results are obtained by examining life expectancy. Source: Calculation by Busetta and Bono [13] based on United Nations, DESA, Population Division. World population prospects: the 2019 revision; https://population.un.org/wpp/Download/Standard/Population (accessed on 1 December 2020).

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
