# Peer review of "How Important Are Genes to Achieve Longevity?"

_ijms, 2022, doi:10.3390/ijms23105635_

Round 1

Reviewer 1 Report

The authors review an interesting topic of current significance: genetics of longevity. To do so, they review current literature on the subject, focusing on the role of genes described to date. It is a paper that is well written in general, that reviews updated and some relevant literature, but that should review some aspects in order to be accepted:

- Starting with the title. The title proposes a question that in my opinion is not answered by the article.

- The origin of the data should be referenced in Table 1. And probably, since the review focuses on genes and longevity, a table with the results obtained in GWAS on genes could also be included, to show how many times each gene has been involved.

- It is clear that during the last few years the rise of GWAS has produced an enormous number of results for all diseases. And longevity has not been left out. Over the last few years a lot of GWAS data have accumulated with promising results, even using alternative phenotypes (paternal longevity, healthspan, or some metaanalysis), which I would like to see included in this review (for instance PMID: 32678081, 30642433, https://doi.org/10.1038/s43587-021-00159-8, among others). And include their results in the discussion.

- Regarding the limitations in point 2.4, the authors again ignore the GWAS that have employed other approaches in the longevity study. Even so, they argue that using unrelated samples is already sufficient in the choice of controls, since controls that will end up living very long can be disregarded. I don't agree with this statement either. It depends on how LLI is defined. The authors themselves, in the article propose that it can be relaxed up to 85 years (line 170). It does not seem so rare to reach 85 years....

- In point 3 the authors discuss 2 genes that have been repeatedly associated with longevity (APOE and FOXO3A) and also DNA repair genes. Here again, I miss a table with all the results to date, because otherwise, the choice of DNA repair genes seems to me to be biased. There are many other pathways related to longevity by various studies (heme metabolism, immunity, homeostasis, wound healing, etc). An explanation of why this pathway is chosen and not others is necessary.

- The authors refer to the evolutionary theory known as "mutation accumulation" without naming it (line 332-onwards), but do not cite the antagonistic pleiotropy theory of aging which should be taken into account.

- Line 347. The concept of LLF is introduced in the last paragraph of the conclusion. The concept, if necessary, should be introduced earlier in the text and the last paragraph should be used to summarize the main idea of the article.

Reviewer 2 Report

In this review, Caruso and coll. tries to provide an overview of the importance of genetic factors in human longevity. The authors seem to believe that gene variants contributing to a long life are mainly those involved with the basic maintenance and function of cells. After a first section in which they describe the methods used to identify potential longevity genes, the authors focus in particular on the APOE, FOXO3A and DNA repair genes, which have been extensively studied in the past and are the only ones found so far in association with human longevity in replication studies.

Comments to the authors

The title of the manuscript somehow fails to deliver on its promises. I would have expected a critical discussion on the (low) importance of genes in human longevity, rather than grabbing the reader's attention with methodologies how to identify them.

Page 2, lines 57-63. The authors criticize the article by Ruby et al., 2018 claiming it estimated only the heritability of lifespan and not of longevity. First, it is known that up until 70 or 80 years of life, lifestyle is a stronger determinant of health and longevity than genetics. Ruby's study had considered the age range 0-120 years so there was no bias due to the lack of data beyond 80 years of age. Secondly, the authors forgot to mention also the article by Kaplanis et al. Science 2018; 360: 171–175 which also gave a low heritability estimate.

Page 2, lines 71-73. I do not understand why the authors, referring to Mazumder's article, attributed the short stature of conscripts only to prenatal exposure to the flu; other factors such as war stress and famine may have been responsible.

Page 2, line 77. The authors seem to adhere to Albert Evans' social gradient theory according to which the longevity of a given population increases in direct proportion to its economic development. However, this belief is disproved by the case of the Okinawan population, as rightly pointed out by Cockerham et al. Soc Sci Med. 2000; 51 (1): 115-22.

Page 3, lines 100-101 and page 5, lines 190-192. Oversimplified seem instead the authors' considerations. Very few genes have been found in association with longevity. As it is known, the presence of “harmful” alleles does not compromise the achievement of longevity. Among the various explanations of this fact are 1) the opposing influence of genes on different health traits; 2) the antagonistic change of gene effect at different ages; 3) the gene – gene interaction (epistasis); and 4) the gene – environment interaction. The authors seem to have mentioned only the latter.

Page 5. Among the study methodologies to identify longevity genes, the authors have forgotten to mention the study of the children of long-lived individuals, who are more likely to stay healthy longer and to live to an older age than their peers.

Page 6, lines 240-241. The study cited in reference no. 33 was clearly not informative as it was based on only two APOE4 positive samples (one in the centenarians and one in nonagenarians). Consequently, any speculation on non-genetic factors that would mitigate the effect of APOE4 is weak.

Additional minor comments:

Although the language of the manuscript is generally clear, it would be advisable to change “centenaries” into “centenarians” (page 5, line 200) and  “Next Generation Sequence” into “Next Generation Sequencing” (page 9, Abbreviations).

Round 2

Reviewer 2 Report

I admit that the authors have made significant changes to their manuscript. I appreciated the addition of the new paragraph 2.4, which mentions the centenarian offspring model. Moreover, they mentioned the presence of detrimental alleles which do not preclude the achievement of longevity. Finally, the authors softened the claim that longevity is more easily achieved in the most prosperous populations. A certain excessive trust in the role of genetic factors on human longevity remains in the text, but it is clearly a legitimate opinion which has its own plausibility that must be respected. I do not think it is necessary to prolong the referral process of this manuscript with further criticisms that would appear pedantic. I report only two typos (on page 10, line 458, “live” instead of “life” and line 461 “consideri”) that should be corrected before further processing the manuscript.